# Himawari-8-derived diurnal variations of ground-level PM₂.₅ pollution across China using the fast space-time Light Gradient Boosting Machine (LightGBM)

Jing Wei[1, 2, 3*], Zhanqing Li[2**], Rachel T. Pinker[2], Jun Wang[3], Lin Sun[4], Wenhao Xue[1], Runze Li[1], Maureen Cribb[2]

1. State Key Laboratory of Remote Sensing Science, College of Global Change and Earth System Science, Beijing Normal University, Beijing, China
2. Earth System Science Interdisciplinary Center, Department of Atmospheric and Oceanic Science, University of Maryland, College Park, MD, USA
3. Department of Chemical and Biochemical Engineering, Iowa Technology Institute, Center for Global and Regional Environmental Research, The University of Iowa, Iowa City, IA, USA
4. College of Geodesy and Geomatics, Shandong University of Science and Technology, Qingdao, China

Correspondence to: Jing Wei (weijing_rs@163.com), Zhanqing Li (zli@atmos.umd.edu)

## Abstract

Information on the particulate matter with a diameter of less than 2.5 μm (PM₂.₅) has been used as an important atmospheric environmental parameter mainly because of its impact on human health. PM₂.₅ is affected by both natural and anthropogenic factors that usually have strong diurnal variations. Such information helps toward understanding the causes of air pollution as well as our adaptation to it. Most existing PM₂.₅ products have been derived from polar-orbiting satellites. This study exploits the use of the next-generation geostationary meteorological satellite Himawari-8/AHI to document the diurnal variation of PM₂.₅. Given the huge volume of satellite data, based on the idea of gradient boosting, a highly efficient tree-based Light Gradient Boosting Machine (LightGBM) method by involving the spatiotemporal characteristics of air pollution, namely the space-time LightGBM (STLG) model, is developed. An hourly PM₂.₅ dataset for China (i.e., ChinaHighPM₂.₅) at a 5-km spatial resolution is

derived based on Himawari-8/AHI aerosol products with additional environmental variables. Hourly $PM_{2.5}$ estimates (number of data samples = 1,415,188) are well correlated with ground measurements in China (cross-validation coefficient of determination, CV-$R^2$ = 0.85), with a root-mean-square error (RMSE) and mean absolute error (MAE) of 13.62 and 8.49 µg/m$^3$, respectively. Our model captures well the $PM_{2.5}$ diurnal variations showing that pollution increases gradually in the morning, reaching a peak at about 10:00 a.m. local time, then decreases steadily until sunset. The proposed approach outperforms most traditional statistical regression and tree-based machine-learning models, with a much lower computation burden in terms of speed and memory, making it most suitable for routine pollution monitoring.

## 1. Introduction

China has faced severe environmental problems during the last two decades, especially air pollution (An et al., 2019; Chan & Yao, 2008; Z. Li et al., 2017; Q. Zhang et al., 2019; Wei et al., 2021a). The sources of air pollution are numerous, coming from both natural changes (e.g., forest fires, biomass burning) and human activities (e.g., industrial production, transportation) (Huang et al., 2014; Sun et al., 2004; Wei et al., 2019a, 2019b, 2021b). Particulate matter with a diameter of less than 2.5 µm ($PM_{2.5}$) has a greater impact on the atmospheric environment and climate change than other air pollutants [e.g., $PM_{10}$, nitrogen dioxide ($NO_2$), and sulfur dioxide ($SO_2$)] (Jacob & Winner, 2009; Z. Li et al., 2017, 2019; Ramanathan & Feng, 2009). Moreover, they can cause great harm to human health due to their smaller particle size (Delfino et al., 2005; Kampa & Castanas, 2008; Kim et al., 2015; Lelieveld et al., 2015).

China has established and operates multiple ground-based observation networks to monitor air pollution in real-time across mainland China, including information about $PM_{2.5}$ pollution.
For near-surface concentrations, the networks provide high-quality $PM_{2.5}$ measurements every hour (even every few minutes) but with non-uniform coverage. In recent years, an increased effort has been made in estimating $PM_{2.5}$ with products generated from multiple instruments on sun-synchronous satellites, e.g., the Multi-angle Imaging SpectroRadiometer (MISR) (Y. Liu et al., 2005; van Donkelaar et al., 2006), the Moderate-resolution Imaging Spectroradiometer (MODIS) (Y. Liu et al., 2007; Ma et al., 2014; Wei et al., 2019a, 2020, 2021a), and the Visible Infrared Imaging Radiometer Suite (VIIRS)

(Wei et al., 2021c; Wu et al., 2016; Yao et al., 2019). However, due to their low revisit cycles (one or two overpasses per day), they are unable to monitor the diurnal variation of pollution. Currently, most

available PM$_{2.5}$ datasets are at low temporal resolutions that cannot meet the requirements of air pollution real-time monitoring (Lennartson et al., 2018). For example, knowing when heavy pollution might occur during the day, people may adjust their time outdoors doing activities accordingly. Following the launch of the Himawari-8/Advanced Himawari Imager (AHI) on 7 October 2014 (Bessho et al., 2016; Letu et al., 2020), near-surface PM$_{2.5}$ concentrations in the Eastern Hemisphere can now be

estimated and used to examine their diurnal cycle.

W. Wang et al. (2017) used the linear mixed-effect (LME) model, and Sun et al. (2019) applied the geographically weighted regression (GWR) and support vector regression (SVR) models to estimate hourly PM$_{2.5}$ concentrations in the Beijing–Tianjin–Hebei (BTH) region from the Himawari-8 aerosol optical depth (AOD) product. T. Zhang et al. (2019) developed an improved LME model, and Xue et al.

(2020) proposed an improved geographically and temporally weighted regression (IGTWR) model to derive hourly PM$_{2.5}$ maps based on the Himawari-8 AOD product over central and eastern China. In addition to traditional statistical regression models, several artificial intelligence models, including the random forest (RF), the gradient boosting decision tree (GBDT), the eXtreme Gradient Boosting (XGBoost), and the deep neural network (DNN), have been recently successfully adopted to obtain

ground-level PM$_{2.5}$ concentrations to local regions and to the whole of China (J. Chen et al., 2019; Gui et al., 2020; J. Liu et al., 2019; Sun et al., 2019; T. Zhang et al., 2020). Nevertheless, due to their poor data-mining ability, traditional statistical regression methods usually suffer from large uncertainties. While artificial intelligence methods can achieve high accuracies, they are often highly demanding on computational power and are thus often slow. Therefore, Spatiotemporal variations of PM$_{2.5}$ have often

been neglected in the models developed in previous studies (J. Chen et al., 2019; J. Liu et al., 2019; Sun et al., 2019; W. Wang et al., 2017; T. Zhang et al., 2019), resulting in relatively low accuracies.

Focusing on the above issues, we have developed a new, highly efficient, and precise method for improving ground-level PM$_{2.5}$ estimates by incorporating spatial and temporal information into the tree-based Light Gradient Boosting Machine (LightGBM) model. This new model is called the space-time

LightGBM (STLG) model, used to generate a high-quality, high-temporal-resolution (hourly) PM$_{2.5}$

dataset over eastern China (at a spatial resolution of 5 km) from the Himawari-8/AHI hourly AOD product. Section 2 provides details about the data used and introduces the development of the STLG model. Section 3 validates the hourly $PM_{2.5}$ estimates and shows the diurnal $PM_{2.5}$ variations across China. Comparisons with results from traditional models and from previous studies are also presented. Section 4 summarizes the study.

## 2.  Materials and methods

### 2.1  Data sources

#### 2.1.1 $PM_{2.5}$ and AOD data

$PM_{2.5}$ hourly measurements from 1583 monitoring stations across China for the year 2018 were collected [Figure 1 in Wei et al. (2020)]. The latest Himawari-8 Version 2 hourly 5-km AODs at 500 nm across mainland China for that year were also collected. This AOD product is synthesized from Level 2 10-minute AODs, generated by a newly developed Lambertian-surface-assumed aerosol retrieval algorithm (Letu et al., 2018; Yoshida et al., 2018). Himawari-8 AOD retrievals have been preliminarily evaluated against in situ AOD retrievals provided by the Aerosol Robotic Network (Giles et al., 2019) and the Sun–Sky Radiometer Observation Network (Z. Li et al., 2018), showing that they are consistent (R = 0.75), with a root-mean-square error (RMSE) and mean absolute error (MAE) of 0.39 and 0.21, respectively (Wei et al., 2019c). Here, only low-uncertainty AOD retrievals (500 nm) were selected for estimating $PM_{2.5}$ concentrations.

#### 2.1.2 Meteorological conditions

$PM_{2.5}$ can be significantly affected by meteorological conditions (Su et al., 2018). However, most currently available reanalysis meteorological products have low temporal resolutions (~3–6 hours). Recently (14 June 2018), the fifth-generation European Centre for Medium-range Weather Forecasts (ECMWF) global atmospheric reanalysis (ERA5) at a horizontal resolution of 0.25°×0.25° has been released, as well as the land version (12 July 2019) at a horizontal resolution of 0.1°×0.1°, both at an hourly time scale (1979 to the present). Here, we use seven ERA5 hourly meteorological parameters,

i.e., the 2-m temperature (TEM), total evaporation (ET), relative humidity (RH), 10-m u- and v-components of wind, surface pressure (SP), and boundary-layer height (BLH).


### 2.1.3 Human influences

Human activity is a key factor affecting $PM_{2.5}$ pollution. The global annual LandScan$^{TM}$ product at a 1-km spatial resolution for the year 2018 was selected to obtain the population distribution (POP) (Dobson et al., 2000). Monthly anthropogenic source emission data from the Multi-resolution Emission

Inventory for China (MEIC) (M. Li et al., 2017; Zheng et al., 2018) were also employed. This dataset is generated from agricultural, industrial, power, residential, and transportation information obtained at more than 700 anthropogenic sources, including a total of 10 atmospheric pollutants and greenhouse gases. Here, four main precursors were selected, i.e., ammonia ($NH_3$), nitrogen oxides ($NO_x$), $SO_2$, and volatile organic compounds (VOC), and direct emissions to PM.


### 2.1.4 Ancillary data

Two additional ancillary datasets, namely, the MODIS monthly Normalized Difference Vegetation Index (NDVI) at a horizontal resolution of $0.05° × 0.05°$ and the Shuttle Radar Topography Mission (SRTM) 90-m digital elevation model (DEM) products, were selected to characterize land cover, its

change and topographical conditions in China. All selected variables (Table 1) with potential impacts on $PM_{2.5}$ concentrations were resampled to the same spatial resolution as the Himawari-8 aerosol product, namely, $0.05° × 0.05°$.

*[Please insert Table 1 here]*

## 2.2 Space-Time LightGBM model

### 2.2.1 LightGBM model

The LightGBM model, a newly developed tree-based machine-learning approach, was introduced in 2017 (Ke et al., 2017). Using the gradient boosting framework to construct the decision tree, this approach can tackle both regression and classification tasks, and as such, can be expanded for PM

applications. It can also tackle the main challenge faced in traditional machine-learning approaches,

namely, computational complexities, which are very time-consuming. LightGBM is a fast, distributed and highly efficient method that reduces the number of data samples ($M$) and features ($N$). The LightGBM model includes three main steps when constructing the decision tree:

1) Histogram-based algorithm. Continuous features are first converted to different bins, which are used to construct feature index histograms without the need to sort during training. It goes through all the

data bins to find the best split point from the feature histograms, which can significantly reduce the computation cost of the split gain. The overall complexity is $O\ (M \times N)$.

2) Gradient-based one-side sampling. Data samples are first sorted in descending order according to their absolute gradients, and the top $a$% of them are selected as a subset sample with large gradients. The $b$% samples are then randomly chosen from the remaining data as a subset sample with small

gradients. The sampled data with small gradients are multiplied by a weight coefficient ($\frac{1-a}{b}$). Consequently, a new classifier is learned and established using the above-sampled data until convergence.

3) Exclusive feature bundling. A graph with weighted edges is first constructed, and each weight corresponds to the total number of conflicts between two features. The features are then sorted in

descending order according to the degree of each feature (the greater the degree, the greater the conflict with other points). Last, each feature is checked in the sorted sequence and either assigned to a combination with small conflicts or created a new combination.

In addition to the main technologies mentioned above, there are other features of the optimization, such as the leaf-wise tree growth strategy with depth restriction (Shi, 2007), histogram difference

acceleration, sequential access gradient, and the support of category feature and parallel learning. These advanced methodologies make it possible to reach a high accuracy and efficiency (Ke et al., 2017).

### 2.2.2 Model development

It is well known that air pollution has spatiotemporal heterogeneity, leading to large differences in

$PM_{2.5}$ concentrations in both time and space. Such characteristics have always been ignored in most traditional statistical regression and artificial intelligence methods. Studies have shown that including

spatiotemporal information has led to improved PM$_{2.5}$ estimates using remote sensing techniques (Z. Li et al., 2017; Wei et al., 2019a, 2020). Therefore, we have introduced a new approach to integrating spatiotemporal information into the LightGBM model. The new model developed here is called the

STLG model. The spatial feature is represented by the geographical distances of one pixel to other points in the circumscribed rectangle of the study region (Baez-Villanueva et al., 2020; Behrens et al., 2018). The distance is calculated using the haversine method (Equation 1) to reflect the spherical distance between two points in the sphere space (Wei et al., 2021a). The temporal feature is represented by the day of the year (DOY), used to distinguish each data record on different days of the year during

the model training.

$$DIS = 2 * r * \text{asin} \left( \sqrt{sin^2 \left( \frac{\varphi_2 - \varphi_1}{2} \right) + \cos(\varphi_1)\cos(\varphi_2) \, sin^2 \left( \frac{\gamma_2 - \gamma_1}{2} \right)} \right) , \quad (1)$$

where $\varphi$ and $\gamma$ represent the latitude and longitude of a point on the sphere, respectively, and $r$ denotes Earth's mean radius ($\approx$ 6371 km). Figure 1 illustrates the flowchart of the new STLG model.

*[Please insert Figure 1 here]*

In addition to Himawari-8 AODs, other auxiliary variables were considered and employed to improve PM$_{2.5}$-AOD relationships. However, to avoid redundant information, we first calculated the normalized importance (%) of each feature to the PM$_{2.5}$ estimation during the model training (Figure 2). It represents the total gains of splits that use the feature during the decision-tree construction, but not the physical contribution. AOD is found to be the most important feature, accounting for about 17%. All

meteorological factors have an important impact on the PM$_{2.5}$ estimation, especially BLH, RH, and TEM (importance > 8%), followed by two surface-related variables (i.e., NDVI and DEM) and POP. The influence of aerosol precursors and emissions (i.e., NH$_3$, NO$_x$, SO$_2$, PM, and VOC) on the PM$_{2.5}$ estimation cannot be ignored (importance > 2%). Therefore, all 16 selected variables are included to establish the final model in this study.

*[Please insert Figure 2 here]*

Here, two independent ten-fold cross-validation methods (10-CV) (Rodriguez et al., 2010), based on all the data samples (i.e., out-of-sample) and PM$_{2.5}$ monitoring stations (i.e., out-of-station), were selected to validate the model performance and the spatial prediction ability, respectively.

## 3. Results and discussion

### 3.1 Model fitting and validation

### 3.1.1 Spatial-scale performance

The STLG model can largely minimize overfitting, showing a strong data-mining ability (Figure 3), which can more accurately establish the relationships between hourly PM$_{2.5}$ observations and influential variables (i.e., coefficient of determination, $R^2 = 0.97$–$0.98$, RMSE = $4.18$–$7.31$ $\mu g/m^3$). Figure 4 illustrates the out-of-sample evaluation results of estimated hourly PM$_{2.5}$ values over China from 08:00 to 17:00 local time in 2018. The STLG model is highly accurate in estimating hourly PM$_{2.5}$ concentrations, with high sample-based CV-R$^2$ values ranging from 0.81 to 0.85, strong slopes of ~0.81–0.84, and small y-intercepts of ~5.52–7.84 $\mu g/m^3$. The uncertainties are overall small, with RMSEs (MAEs) ranging from 11.24 (6.82) $\mu g/m^3$ to 15.56 (9.79) $\mu g/m^3$. However, the STLG performs slightly differently, with small differences in main evaluation indicators throughout the day. The main reason being that the number of training samples is reduced during sunrise (Figure 4a-b) and sunset (Figure 4i-j) in optical remote sensing, affecting the model training. Air pollution also has clear diurnal variations at different PM$_{2.5}$ pollution levels due to the different intensities of human activities and natural conditions. In general, our model is stable and robust, with an equal out-of-sample CV-R$^2$ of 0.85 and an equal regression slope of 0.81 at most hours during the day in China (Figure 4c-h).

*[Please insert Figures 3 and 4 here]*

Furthermore, out-of-station CV-R$^2$ values range from 0.76 to 0.81, and RMSE (MAE) values range from 12.49 (7.85) $\mu g/m^3$ to 17.61 (11.33) $\mu g/m^3$ (Figure 5), indicating that our model has a strong spatial prediction ability and can well predict PM$_{2.5}$ values in those areas without surface observations.. The station-based accuracy is also slightly decreased with reference to the sample-based accuracy,

further illustrating the robustness of our model. However, two cross-validation results (e.g., slopes = 0.78–0.84) indicate that hourly PM$_{2.5}$ concentrations are overall underestimated (Figures 4–5), a common issue in fine-particle remote sensing (Wei et al., 2020). This can be explained by the large

aerosol retrieval uncertainty, as well as the small number of data samples under highly polluted conditions (Wei et al., 2019c, d).

*[Please insert Figure 5 here]*

Evaluated was also the regional performance of the STLG model for hourly PM$_{2.5}$ estimates (Figure 6). Hourly PM$_{2.5}$ estimates (number of data samples, N = 1,151,595) are highly consistent with ground

measurements, with a high sample-based CV-R$^2$ of 0.87 and a strong regression slope of 0.86, showing small estimation uncertainties (i.e., RMSE = 12.77 μg/m$^3$, MAE = 8.12 μg/m$^3$) over Eastern China. The STLG model performs well (e.g., CV-R$^2$ = 0.88, slope = 0.87) in two typical urban agglomerations of public concern in China, i.e., the Beijing-Tianjin-Hebei (BTH) (Figure 6b) and Yangtze River Delta (YRD) (Figure 6c) regions. By contrast, our model performs relatively poorly in the Pearl River Delta

(PRD) region (Figure 6d), possibly due to the significant reduction in the number of data samples caused by frequent, long-term cloud cover in southern China. Note that there are some differences in the uncertainty of hourly PM$_{2.5}$ estimates mainly because of varying levels of air pollution. The pollution level in the BTH region is about three times higher than that in the PRD region.

*[Please insert Figure 6 here]*

Figure 7 shows the accuracy of the STLG model at each monitoring station across China. At the individual-site scale, the number of data samples gradually decreases from northern China to southern China, mainly due to increasing cloud contamination, with a site average of 997 data samples in China. Except for several scattered monitoring stations in western China, the STLG model has a high performance and adaptability and can well estimate hourly PM$_{2.5}$ concentrations at most monitoring

stations (e.g., average CV-R$^2$ = 0.78, RMSE = 12.21 μg/m$^3$, and MAE = 8.17 μg/m$^3$). In general, approximately 76%, 79%, and 82% of monitoring stations show high accuracy, with out-of-sample CV-

$R^2$ values > 0.7, RMSE values < 15 μg/m$^3$, and MAE values < 10 μg/m$^3$ in hourly PM$_{2.5}$ estimates, especially for those located in central and northern China.

*[Please insert Figure 7 here]*

**3.1.2 Temporal-scale performance**

We first quantified the time series of the bias in hourly PM$_{2.5}$ estimates during the day in China (Figure 8). There is a slight temporal dependence, where the PM$_{2.5}$ bias increases gradually with increasing standard deviation, reaching a maximum around 11:00 a.m. and subsequently decreasing. This seems to be closely related to the diurnal variation of PM$_{2.5}$ concentrations. The PM$_{2.5}$ estimates are less affected

by the time-dependent bias in the Himawari-8 AOD product (Wei et al., 2019c) because machine learning is not sensitive to the systematic bias of aerosol retrievals (Wei et al., 2021c). Nevertheless, our model is generally robust and can accurately estimate PM$_{2.5}$ concentrations with small mean (median) biases of 0.05–0.08 (0.63–0.99) μg/m$^3$ during different hours throughout the day.

*[Please insert Figure 8 here]*

We also compared Himawari-8-derived and ground-based PM$_{2.5}$ diurnal variations from all available monitoring stations in China and three typical urban clusters (Figure 9). Hourly PM$_{2.5}$ concentrations observed by satellite are highly consistent with ground-based measurements, with a small difference within ±0.10, 0.11, 0.13, and 0.11 μg/m$^3$ in China and in each region, respectively. Moreover, the same diurnal variations of PM$_{2.5}$ pollution are seen during the day, i.e., they reach their maximum values at

10:00 or 11:00 and are lower at sunrise and sunset. These results illustrate that the diurnal PM$_{2.5}$ variations derived from Himawari-8 are reasonable compared to ground-based measurements.

*[Please insert Figure 9 here]*

We investigated the time series of the daily performance of the STLG model in estimating hourly PM$_{2.5}$ concentrations in China. The number of data samples varies on a daily basis, with an average of 3975

per day and with more than 83% of all days having more than 2000 (Figure 10). The large gap in the number of data samples is mainly caused by different degrees of cloud contamination in the satellite

aerosol products for different days. The STLG model captures well the hourly PM$_{2.5}$ values on most days, with an average out-of-sample R$^2$ of 0.73 and average RMSE and MAE values of 13.06 μg/m$^3$ and 8.53 μg/m$^3$, respectively. In general, hourly PM$_{2.5}$ estimates are more reliable on approximately 79% (CV-R$^2$ > 0.7), 70% (RMSE < 15 μg/m$^3$), and 74% (MAE < 10 μg/m$^3$) of the days in the year. The model performance also varies greatly at the seasonal level, with average CV-R$^2$ values of 0.82, 0.71, 0.87, and 0.86, and average RMSE values of 14.55, 9.63, 11.83, and 17.57 μg/m$^3$ in spring, summer, autumn, and winter, respectively (Figure 11). In general, the overall uncertainty of PM$_{2.5}$ estimates increases at the beginning and at the end of the year, likely due to the harsher environmental conditions (e.g., low humidity and less precipitation) and more intense human activities (e.g., coal heating and straw burning) in winter and spring.

*[Please insert Figures 10 and 11 here]*

We have evaluated temporally synthesized PM$_{2.5}$ data from the hourly data samples at each monitoring station for the year 2018 (Figure 12). Daily mean PM$_{2.5}$ estimates are highly correlated to those calculated from surface observations (R$^2$ = 0.91), and the average RMSE (MAE) value is 10.11 (6.39) μg/m$^3$. This suggests that the STLG model can capture daily PM$_{2.5}$ variations more accurately. Note that daily synthetic PM$_{2.5}$ data derived from geostationary satellites have a higher temporal frequency than data derived from sun-synchronous satellites. In general, PM$_{2.5}$ synthetic values also have high accuracies and low estimation uncertainties (e.g., R$^2$ = 0.98, RMSE = 1.6–3.3 μg/m$^3$, MAE = 1.1–2.3 μg/m$^3$) from monthly to annual scales, allowing for a better description of spatiotemporal distributions and variations of PM$_{2.5}$ pollution across China.

*[Please insert Figure 12 here]*

### 3.2 Spatiotemporal characteristics
### 3.2.1 Diurnal variations
Figure 13 shows Himawari-8-derived hourly mean near-surface PM$_{2.5}$ concentrations from 08:00 to 17:00 local time in 2018 across mainland China. They do not cover western Xinjiang and Tibet due to the limitation of satellite scanning. PM$_{2.5}$ pollution varies diurnally across China, being at an overall low

level at sunrise (~29.94±10.91 μg/m$^3$). With the increase in human activities, air pollution becomes more severe over time, reaching a peak at around 10:00–11:00 local time in China (~36±13 μg/m$^3$). These high levels of pollution can last several hours. As the day progresses, human activities subside, and atmospheric fine particles settle on surfaces. PM$_{2.5}$ concentrations thus decrease towards sunset in most areas in China (~23.21±9.73 μg/m$^3$). In general, air pollution in the morning (i.e., 08:00–12:00) is much more severe than in the afternoon (i.e., 13:00–17:00) in China, with morning PM2.5 concentrations about 1.3 times higher than afternoon levels. This is related to the influence of varying BLHs (Z. Li et al., 2017; Su et al., 2018).

*[Please insert Figure 13 here]*

Table 2 summarizes the diurnal PM$_{2.5}$ variations in eastern China and three typical urban agglomerations. PM2.5 pollution levels in eastern China are generally higher than the national level at each hour of the day due to the dense human population and intensive human activities. In the BTH region, PM2.5 pollution varies greatly, with hourly PM2.5 concentrations ranging from 28.88±10.16 μg/m$^3$ (10:00) to 49.31±15.03 μg/m$^3$ (16:00) and with differences exceeding 20 μg/m$^3$. PM2.5 pollution remained at a high level (> 42 μg/m$^3$) before 12:00 and dropped to a lower level (< 29 μg/m$^3$) after 16:00. This is closely related to people's daily activities and the production and life cycle of PM2.5 during the day, as well as the change of boundary mixing as a function of the day (Lennartson et al., 2018; Wang and Christopher, 2003). Similar patterns and PM2.5 pollution levels are seen in the YRD region. In general, the PRD region is less polluted in the morning but more severely polluted in the afternoon than the BTH region. Compared with the BTH and PRD regions, PM2.5 pollution in the PRD region is much lower and shows a smaller diurnal difference, with hourly PM2.5 values ranging from 29.49±5.97 μg/m$^3$ (11:00) to 36.36±5.76 μg/m$^3$ (08:00). Better natural conditions and fewer pollutant emissions mainly explain this (Su et al., 2018).

In general, our satellite-derived diurnal variations of PM2.5 pollution agree well with ground-based observations at both national and regional levels but with generally lower PM2.5 concentrations (Figure 9). The reason is that the PM2.5 monitoring stations are unevenly distributed and vary greatly in the number of stations at the regional scale. Also, most sites are distributed in urban areas, leading to

inevitable overestimations due to urban-rural differences. However, satellite remote sensing can cope with this deficiency by generating spatially continuous PM$_{2.5}$ maps, providing more accurate information about the distribution and variations of PM$_{2.5}$ pollution.

*[Please insert Table 2 here]*

### 3.2.2 Seasonal and annual variations

Seasonal PM$_{2.5}$ maps are synthesized from daily PM$_{2.5}$ maps from 2018 across China according to our previous approach (Wei et al., 2019a). Our results illustrate that PM$_{2.5}$ pollution varies greatly on a seasonal scale (Figure 14). Pollution levels are generally low and show similar spatial patterns in summer (~22.86±7.05 μg/m$^3$) and autumn (~23.76±10.97 μg/m$^3$) across China (Table 3). By contrast, it is much more severe in spring (~32.84±11.49 μg/m$^3$) and winter (~39.04±16.32 μg/m$^3$) across China, especially in the BTH and YRD regions in winter. The main reasons are the frequent sandstorms and the long-distance transmission of sand and dust in spring, and the burning of coal and fossil fuels for heating in winter, leading to more pollutant emissions in northern China.

*[Please insert Figure 14 and Table 3 here]*

PM$_{2.5}$ pollution also shows significant spatial heterogeneities across China (Figure 15), with an annual mean PM$_{2.5}$ concentration of 28.99±10.31 μg/m$^3$ in 2018 (Table 3). High pollution levels are always observed in the Hebei, Shandong, Jiangsu, Anhui, Henan, Hubei, and Sichuan provinces. Interactions between intensive human activities, adverse stagnant weather (e.g., low BLHs and low winds), and special terrain (e.g., basin) can increase anthropogenic aerosols (Z. Chen et al., 2008; X. Wang et al., 2018). By contrast, PM$_{2.5}$ pollution is relatively light in the northeast (e.g., Heilongjiang and Jilin provinces), the southwest (e.g., Tibet and Yunan provinces), and the eastern coastal areas of China (e.g., Zhejiang and Fujian provinces). These provinces are sparsely populated or experience meteorological conditions favorable for dispersing pollution (Su et al., 2018).

*[Please insert Figure 15 here]*

### 3.3 Discussion

### 3.3.1 Comparison with traditional models

We first compared results from the STLG model with results from five widely used statistical regression models employed for estimating $PM_{2.5}$ in China using the same input dataset (Table 4). The multivariate linear regression (MLR) model performs the worst due to the complex nonlinear $PM_{2.5}$-AOD relationship. The GWR model performs better because it takes into account the spatial characteristics of $PM_{2.5}$ pollution. The generalized additive model (GAM) and the LME model show overall improved performances, with decreasing estimation uncertainties because of their nonlinear characteristics and stronger data regression abilities. The two-stage model outperforms the GAM and MLE models, with higher CV-$R^2$ values and smaller estimation uncertainties, by combining the advantages of the GWR and LME models. Our model performs better than all of the traditional statistical regression models considered, mainly due to its stronger data-mining ability.

*[Please insert Table 4 here]*

The first six rows of Table 5 show the accuracies and efficiencies of six tree-based machine-learning models when estimating $PM_{2.5}$ in China using the same input dataset. The Decision Tree (DT; Quinlan, 1986) is a traditional, frequently used, supervised learning classification method. Although the training speed is the fastest, and the memory consumption is the least, it has the worst performance because of the simple single classifier. The model performances of ensemble-learning approaches, i.e., GBDT (Friedman, 2001), RF (Breiman, 2001), extremely randomized trees (ERT; Geurts et al., 2006), and XGBoost (Chen & Guestrin, 2016), can be significantly improved by combining several weak classifiers into a strong classifier. Among them, the ERT model yields a higher estimation accuracy and a stronger spatial prediction ability than other ensemble-learning models. The LightGBM model (Ke et al., 2017) performs the best, with the highest accuracy and smallest uncertainty among all tree-based machine-learning approaches considered.

*[Please insert Table 5 here]*

The model efficiency differs among these models due to the large differences in the algorithm design frameworks. These tree-based, machine-learning models can be divided into two categories. The DT,

RF, and ERT models fall into the "bagging" category, which synthesizes multiple independent and unrelated weak classifiers into a strong classifier. It allows for work in parallel, which can save much time but may need more computer memory. The GBDT, XGBoost, and LightGBM models fall into the "boosting" category, which synthesizes multiple interdependent and related weak classifiers into a strong classifier. They can only work in serial, which may take much time but not too much memory. In general, the STGB model is the most time-consuming, while the STET model is the most memory-consuming. By contrast, the LightGBM model runs very fast and consumes very little computer memory, benefiting from a series of algorithm optimizations (Ke et al., 2017).

After considering spatiotemporal variations, all the newly defined space-time tree-based machine-learning approaches (i.e., STDT, STGB, STXB, STRF, STET, and STLG) show significant improvements in both overall estimation accuracy and spatial prediction ability in estimating hourly $PM_{2.5}$ concentrations with reference to their original models. This further illustrates the importance of including spatiotemporal information when constructing $PM_{2.5}$–AOD relationships. More importantly, the training speed of these models did not decrease much, and the memory consumption did not increase much either. In general, the STLG model shows the best performance with a high efficiency (i.e., training speed = 46 s, memory usage = 0.60 GB) among all the space-time, tree-based machine-learning models. Therefore, our new STLG model is highly valuable for accurate and fast air pollution monitoring, in particular for our future study extended to the global scale.

### 3.3.2 Comparison with related studies

We compared Himawari-8-based hourly $PM_{2.5}$ estimates at regional and national scales in China with previous related studies (Table 6). Local hourly $PM_{2.5}$ concentrations retrieved from our national-scale model are more accurate than those derived from the models developed separately in local areas, e.g., the LME model (W. Wang et al., 2017), the GWR, SVR, RF, and DNN models in the BTH region (Sun et al., 2019), and the two-stage RF and DNN models in the YRD region (Fan et al., 2020; Tang et al., 2019). Our model also outperforms most of the statistical regression models and machine-learning models focused on the entirety of China, e.g., the I-LME, IGTWR, RF, Adaboost, XGBoost, and their stacked models in China (J. Chen et al., 2019; Liu et al., 2019; Xue et al., 2020; T. Zhang et al., 2019).

This is due to the stronger data-mining ability, considering key spatial and temporal information about air pollution (ignored in previous studies), and introduces more comprehensive factors that affect $PM_{2.5}$ pollution (e.g., emission inventories).

*[Please insert Table 6 here]*

## 4. Summary and conclusion

$PM_{2.5}$ has a great impact on the atmospheric environment and is also used as a key indicator in environmental health studies. It varies diurnally, affected by both natural and human factors. Previous studies have been based on data from sun-synchronous satellites, which can monitor air pollution at coarse temporal scales (i.e., daily) while high-temporal-resolution and accurate information on $PM_{2.5}$ are needed. In this study, the Himawari-8/AHI hourly AOD product is employed to address this issue. Moreover, considering the large volume of input data and the large errors in $PM_{2.5}$ estimation using traditional methods, an efficient and accurate space-time Light Gradient Boosting Machine (i.e., STLG) model has been developed. It utilizes meteorological, human, land use, and topographical parameters and is implemented at 5-km resolution and hourly time scale to generate $PM_{2.5}$ information over China. The hourly $PM_{2.5}$ estimates are evaluated against surface observations, and $PM_{2.5}$ spatiotemporal variations are also investigated.

The STLG model predicts hourly $PM_{2.5}$ values accurately, with high out-of-sample (out-of-station) CV-$R^2$ values of ~0.81–0.85 (~0.76–0.81) and low RMSE values of ~11.24–15.56 (~12.49–17.61) μg/m$^3$ throughout the day. The model can also produce daily (e.g., $R^2 = 0.91$, RMSE = 10.11 μg/m$^3$), monthly, seasonal, and annual mean $PM_{2.5}$ values (e.g., $R^2 = 0.98$, RMSE = 1.6–3.3 μg/m$^3$). $PM_{2.5}$ varies diurnally in most areas of mainland China, where $PM_{2.5}$ concentrations reach a maximum at 10 a.m. and are generally low at sunrise and sunset on a given day. $PM_{2.5}$ also varies greatly on a seasonal basis, where winter and summer experience the highest and lowest air pollution levels, respectively. Comparison results suggest that the proposed model is more accurate than traditional statistical regression models, other tree-based machine learning models, and various models developed in previous studies. Overall, the STLG model is more efficient, with faster training speed and less memory

consumption. These results illustrate that this algorithm can be useful for real-time monitoring of $PM_{2.5}$ pollution in China.

## Data availability

PM$_{2.5}$ measurements are available at http://www.cnemc.cn, the Himawari-8 AOD product is available at ftp.ptree.jaxa.jp, ERA5 reanalysis products are available at https://cds.climate.copernicus.eu/, the MODIS product is available at https://search.earthdata.nasa.gov/, and the LandScan$^{TM}$ product is available at https://landscan.ornl.gov/. The ChinaHighPM$_{2.5}$ dataset is available at https://weijing-rs.github.io/product.html.

## Author contribution

JW designed the research and wrote the initial draft of this manuscript. ZL, RP, JW, and LS reviewed and edited the paper. RL and WX helped to process the data. MC copyedited the article. All authors made substantial contributions to this work.

## Competing interests

The authors declare that they have no conflict of interest.

## Acknowledgements

We would like to thank Dr. Qiang Zhang at Tsinghua University for providing MEIC pollution emission data for China.

## Financial support

This research has been supported by the National Key R&D Program of China (2017YFC1501702) and the National Natural Science Foundation of China (42030606).

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

 **Tables**

**Table 1.** Summary of datasets and sources used in this study.

| Dataset | Variable | Content | Unit | Spatial Resolution | Temporal Resolution | Data Source |
|---|---|---|---|---|---|---|
| $PM_{2.5}$ | $PM_{2.5}$ | $PM_{2.5}$ | $\mu g/m^3$ | in situ | Hourly | CNEMC |
| AOD | AOD | Himawari-8 AOD | - | 5 km × 5 km | Hourly | Himawari-8 |
| Meteorology | ET | Total evaporation | mm | 0.1°×0.1° | Hourly | ERA5 |
|  | SP | Surface pressure | hPa | 0.1°×0.1° |  |  |
|  | TEM | 2-m temperature | K | 0.1°×0.1° |  |  |
|  | WU | 10-m u-component of wind | m/s | 0.1°×0.1° |  |  |
|  | WV | 10-m v-component of wind | m/s | 0.1°×0.1° |  |  |
|  | BLH | Boundary-layer height | m | 0.25°×0.25° |  |  |
|  | RH | Relative humidity | % | 0.25°×0.25° |  |  |
| Emissions | $NH_3$ | Ammonia | Mg/grid | 0.25°×0.25° | Monthly | MEIC |
|  | $NO_x$ | Nitrogen oxides | Mg/grid |  |  |  |
|  | $SO_2$ | Sulfur dioxide | Mg/grid |  |  |  |
|  | VOC | Volatile organic compounds | Mg/grid |  |  |  |
|  | PM | PM, coarse | Mg/grid |  |  |  |
| Land cover | NDVI | NDVI | - | 0.05°×0.05° | Monthly | MOD13C2 |
| Topography | DEM | Surface elevation | m | 90 m × 90 m | - | SRTM |
| Population | POP | Ambient population | - | 1 km × 1 km | Yearly | LandScan™ |

**Table 2**. Hourly mean PM$_{2.5}$ concentrations ($\mu$g/m$^3$) in 2018 in China, eastern China (ECHN), the
Beijing-Tianjin-Hebei (BTH) region, the Yangtze River Delta (YRD), and the Pearl River Delta (PRD).

| Time | China | ECHN | BTH | YRD | PRD |
|---|---|---|---|---|---|
| 08:00 | 29.94±10.91 | 31.97±11.55 | 42.46±12.97 | 38.60±10.57 | 29.34±5.01 |
| 09:00 | 33.37±12.59 | 36.29±13.52 | 47.32±15.04 | 43.55±11.27 | 34.81±5.46 |
| 10:00 | 35.67±13.53 | 38.56±14.05 | 49.31±15.03 | 44.72±11.17 | 35.48±5.47 |
| 11:00 | 35.63±13.05 | 38.72±13.53 | 49.10±13.77 | 44.27±10.55 | 36.36±5.76 |
| 12:00 | 31.23±11.74 | 35.10±12.47 | 42.38±12.86 | 41.37±9.77 | 34.56±5.72 |
| 13:00 | 28.45±11.40 | 32.23±11.73 | 37.70±11.55 | 39.36±9.22 | 33.33±5.48 |
| 14:00 | 26.36±11.18 | 30.14±11.09 | 34.32±11.81 | 37.31±8.59 | 32.05±5.50 |
| 15:00 | 24.25±10.06 | 28.67±10.21 | 31.95±11.26 | 36.77±8.13 | 30.34±5.43 |
| 16:00 | 23.63±9.26 | 27.38±9.15 | 29.82±10.13 | 32.84±6.30 | 29.49±5.97 |
| 17:00 | 23.21±9.73 | 26.63±8.93 | 28.88±10.16 | 27.59±4.39 | 31.56±6.17 |
| Morning | 33.29±11.59 | 36.15±12.41 | 46.12±13.29 | 42.50±10.22 | 34.52±4.63 |
| Afternoon | 25.11±9.78 | 29.01±9.70 | 32.53±10.53 | 34.76±6.66 | 31.42±4.85 |

**Table 3.** Annual and seasonal mean PM$_{2.5}$ concentrations (μg/m$^3$) in 2018 in China, eastern China (ECHN), the Beijing-Tianjin-Hebei (BTH) region, the Yangtze River Delta (YRD), and the Pearl River Delta (PRD).

| Time | China | ECHN | BTH | YRD | PRD |
|------|-------|------|-----|-----|-----|
| Spring | 32.84±11.49 | 34.93±10.95 | 45.75±12.96 | 40.35±9.55 | 33.97±4.50 |
| Summer | 22.86±7.05 | 24.16±6.29 | 29.99±7.46 | 26.16±4.58 | 23.56±3.18 |
| Autumn | 23.76±10.97 | 28.64±11.60 | 35.98±11.20 | 35.97±7.80 | 29.54±4.43 |
| Winter | 39.04±16.32 | 48.34±17.47 | 48.36±18.92 | 57.41±16.88 | 43.92±8.56 |
| Annual | 28.99±10.31 | 32.56±10.78 | 39.32±11.74 | 38.64±8.27 | 32.98±4.53 |


**Table 4.** Comparison of the model performances of widely used models and the STLG model in estimating PM$_{2.5}$ from Himawari-8 data at 14:00 local time in 2018 in China (N = 162,840).

| Model | Out-of-sample validation | | | Out-of-station validation | | |
|---|---|---|---|---|---|---|
| | CV-R$^2$ | RMSE | MAE | CV-R$^2$ | RMSE | MAE |
| MLR | 0.19 | 24.17 | 22.89 | 0.19 | 24.19 | 22.91 |
| GWR | 0.39 | 21.96 | 20.74 | 0.37 | 22.42 | 21.02 |
| GAM | 0.39 | 19.09 | 18.64 | 0.36 | 19.77 | 18.89 |
| LME | 0.50 | 18.91 | 17.34 | 0.48 | 19.06 | 17.95 |
| Two-stage | 0.58 | 17.60 | 15.71 | 0.54 | 17.99 | 16.01 |
| STLG | 0.85 | 13.09 | 8.11 | 0.81 | 14.63 | 9.29 |

**Table 5.** Comparison of the model performances of different tree-based machine-learning models and the STLG model using the same input data. Data are from 14:00 local time in 2018 in China (N = 162,840).

| Model | Out-of-sample validation | | | Out-of-station validation | | | Speed (s) | Memory (GB) |
|---|---|---|---|---|---|---|---|---|
| | $R^2$ | RMSE | MAE | $R^2$ | RMSE | MAE | | |
| DT | 0.52 | 25.53 | 14.80 | 0.48 | 27.03 | 15.57 | 6 | 0.58 |
| GBDT | 0.65 | 20.03 | 13.17 | 0.61 | 21.20 | 14.10 | 94 | 0.59 |
| XGBoost | 0.73 | 17.94 | 10.78 | 0.68 | 19.59 | 11.93 | 456 | 0.69 |
| RF | 0.72 | 17.86 | 11.33 | 0.69 | 18.80 | 11.95 | 165 | 2.59 |
| ERT | 0.74 | 17.12 | 10.87 | 0.72 | 18.01 | 11.49 | 54 | 3.69 |
| LightGBM | 0.78 | 15.79 | 9.84 | 0.73 | 17.59 | 11.21 | 34 | 0.60 |
| STDT | 0.65 | 21.09 | 12.33 | 0.63 | 22.00 | 12.85 | 8 | 0.60 |
| STGB | 0.75 | 16.82 | 10.93 | 0.73 | 17.61 | 11.54 | 503 | 0.61 |
| STXB | 0.82 | 14.73 | 8.76 | 0.78 | 15.92 | 9.62 | 456 | 0.68 |
| STRF | 0.81 | 14.62 | 9.17 | 0.79 | 15.44 | 9.69 | 219 | 2.75 |
| STET | 0.82 | 14.42 | 8.95 | 0.80 | 15.30 | 9.55 | 77 | 4.25 |
| STLG | 0.85 | 13.09 | 8.11 | 0.81 | 14.63 | 9.29 | 46 | 0.60 |

**Table 6.** Comparison of model performances from previous studies in estimating hourly PM$_{2.5}$ concentrations in China.

| Model | Model validation | | | Region | Reference |
|---|---|---|---|---|---|
| | $R^2$ | RMSE | MAE | | |
| LME | 0.86 | 24.5 | 14.2 | BTH | W. Wang et al. (2017) |
| LME | 0.63 | 29.0 | 18.1 | BTH | Sun et al. (2019) |
| GWR | 0.76 | 23.3 | 16.7 | | Sun et al. (2019) |
| SVR | 0.77 | 21.5 | 12.3 | | Sun et al. (2019) |
| RF | 0.82 | 20.3 | 12.1 | | Sun et al. (2019) |
| DNN | 0.84 | 19.9 | 11.9 | | Sun et al. (2019) |
| two-stage RF | 0.86 | 12.4 | - | YRD | Tang et al. (2019) |
| DNN | 0.86 | 14.3 | - | YRD | Fan et al. (2020) |
| RF | 0.82 | 19.6 | 12.2 | China | J. Chen et al. (2019) |
| Adaboost | 0.84 | 18.3 | 10.7 | | J. Chen et al. (2019) |
| XGBoost | 0.84 | 18.1 | 11.4 | | J. Chen et al. (2019) |
| Stacked model | 0.85 | 17.3 | 10.5 | | J. Chen et al. (2019) |
| RF | 0.86 | 17.3 | 10.3 | China | Liu et al. (2019) |
| I-LME | 0.84 | - | - | BTH | T. Zhang et al. (2019) |
| | 0.80 | - | - | YRD | |
| | 0.74 | - | - | PRD | |
| | 0.82 | - | - | China | |
| IGTWR | 0.78 | 21.1 | - | China | Xue et al. (2020) |

**Figures**

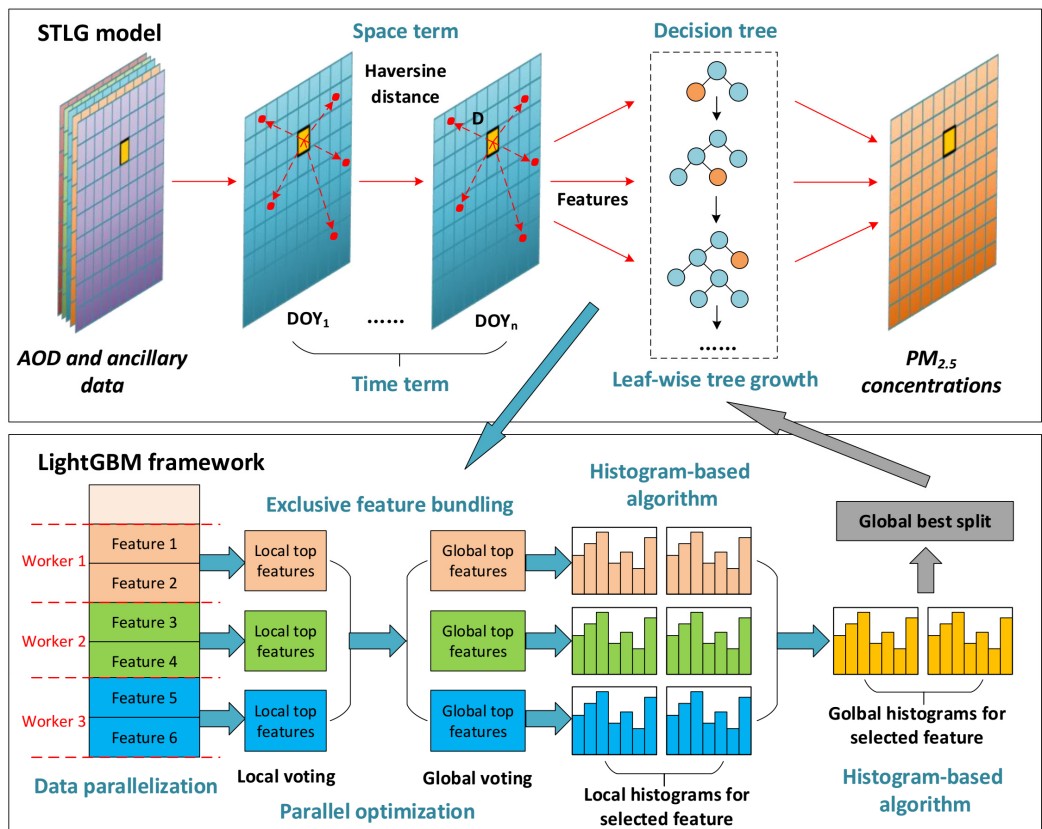

 **Figure 1.** Schematics of the space-time LightGBM (STLG) model developed in this study (upper panel) and the framework of the original LightGBM model (bottom panel).

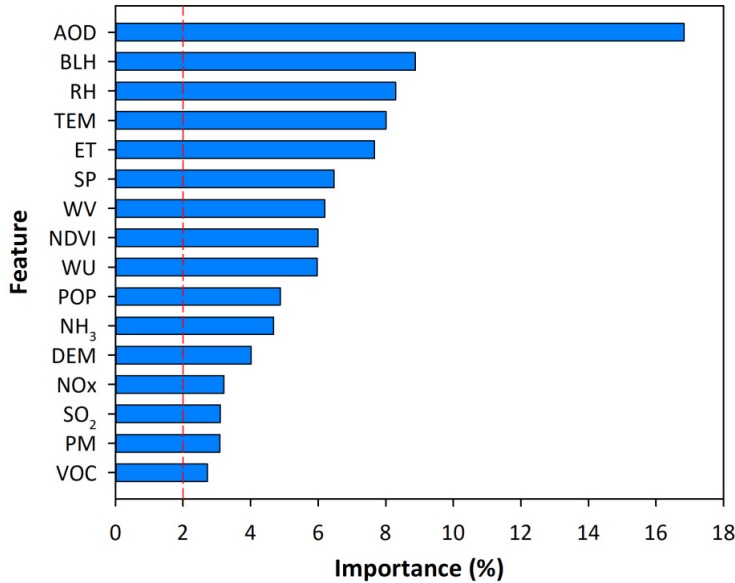

**Figure 2.** Sorted normalized importance (%) of each feature in the PM$_{2.5}$ estimation during the model construction.


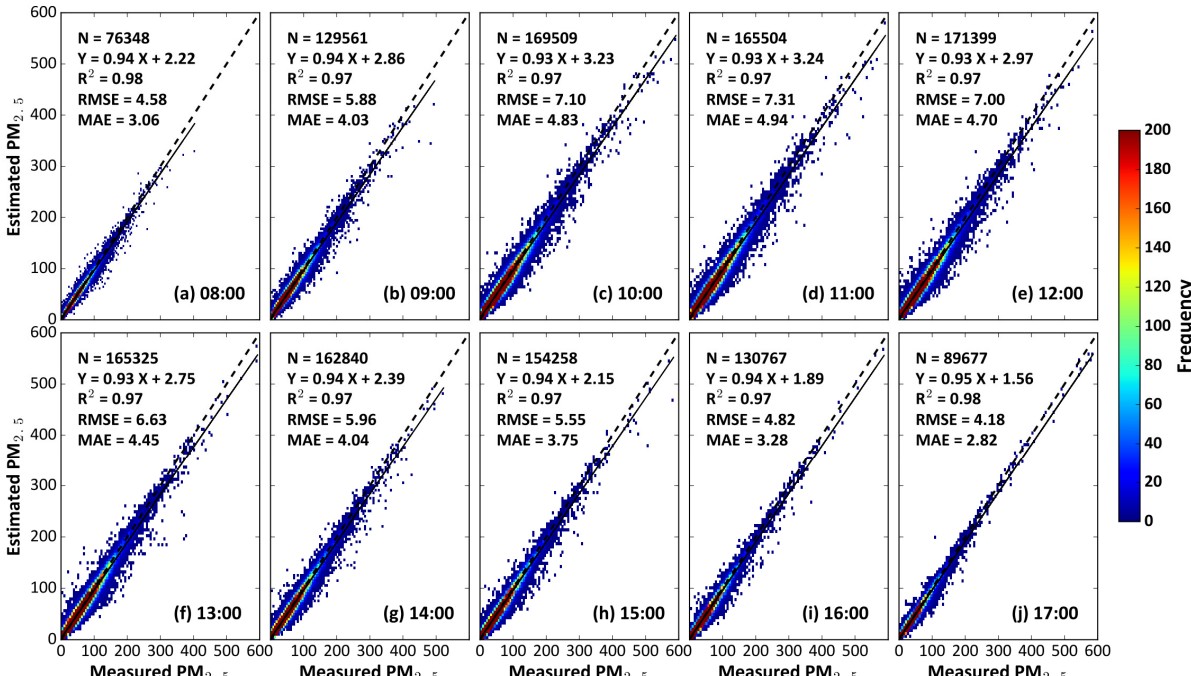

**Figure 3.** Density scatterplots of model-fitted PM2.5 estimates (μg/m³) at (a) 08:00, (b) 09:00, (c) 10:00, (d) 11:00, (e) 12:00, (f) 13:00, (g) 14:00, (h) 15:00, (i) 16:00, and (j) 17:00 local time in 2018 in China. Dashed lines denote 1:1 lines, and solid lines denote best-fit lines from linear regression.


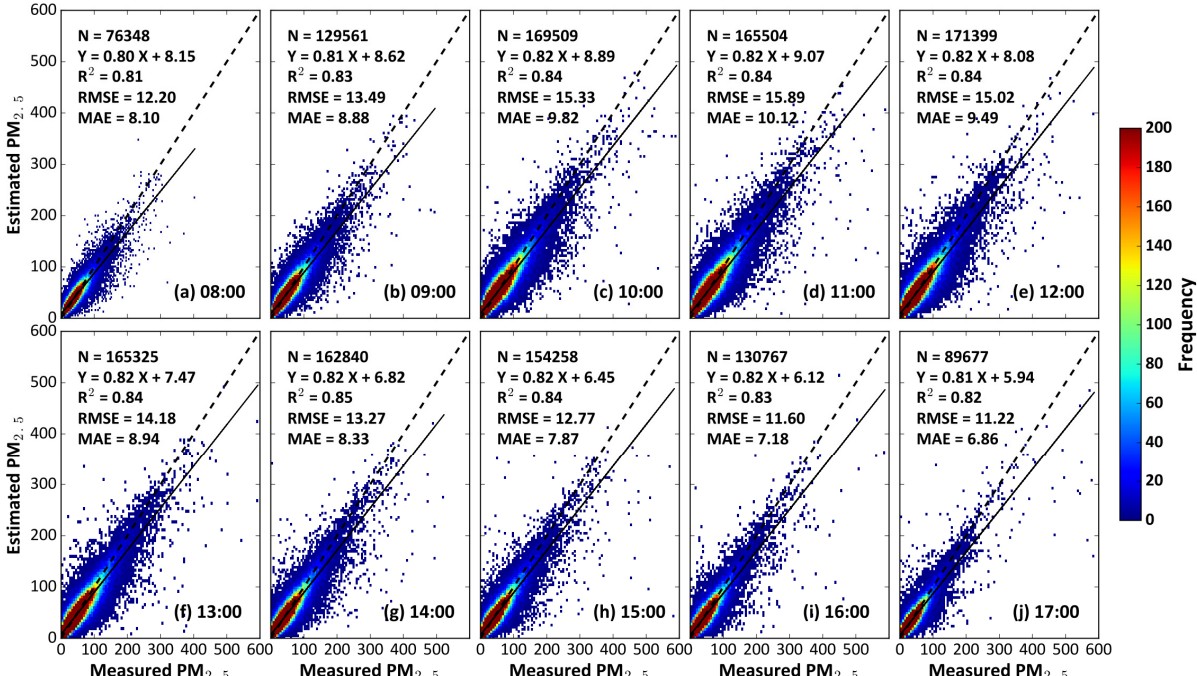

**Figure 4.** Density scatterplots of out-of-sample cross-validation results of PM$_{2.5}$ estimates (μg/m$^3$) at (a) 08:00, (b) 09:00, (c) 10:00, (d) 11:00, (e) 12:00, (f) 13:00, (g) 14:00, (h) 15:00, (i) 16:00, and (j) 17:00 local time in 2018 in China. Dashed lines denote 1:1 lines, and solid lines denote best-fit lines from linear regression.


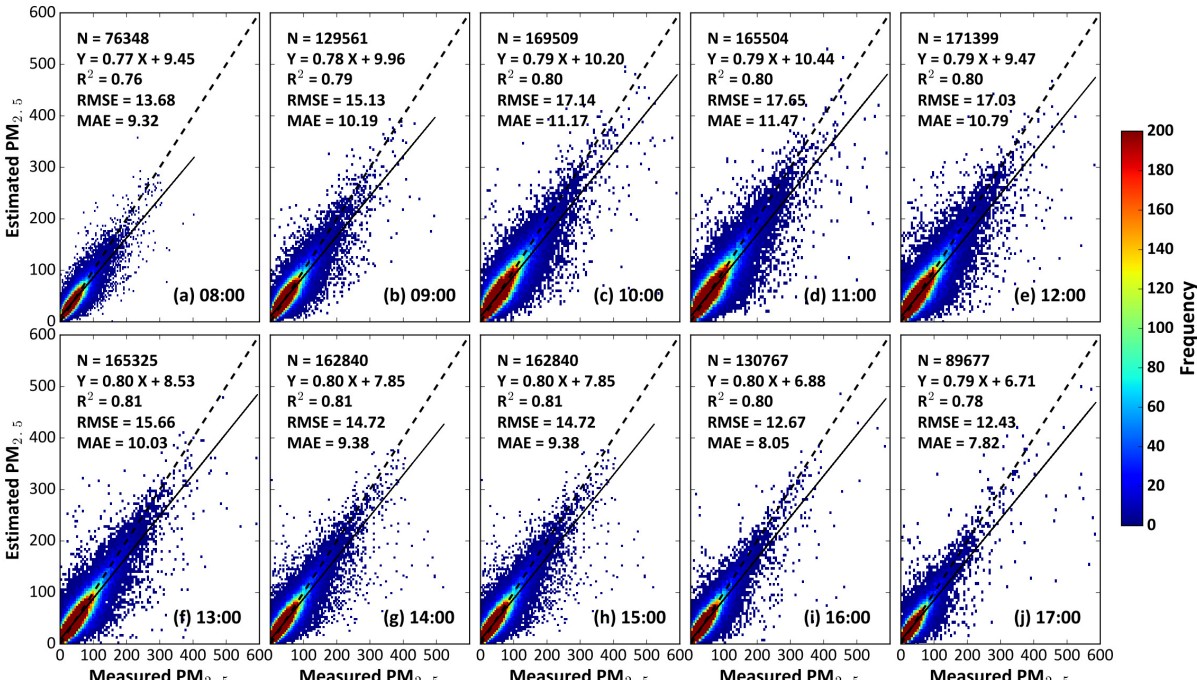

**Figure 5.** Density scatterplots of out-of-station cross-validation results of PM2.5 estimates (μg/m$^3$) at (a) 08:00, (b) 09:00, (c) 10:00, (d) 11:00, (e) 12:00, (f) 13:00, (g) 14:00, (h) 15:00, (i) 16:00, and (j) 17:00 local time in 2018 in China. Dashed and solid lines denote 1:1 and best-fit lines from linear regression, respectively.

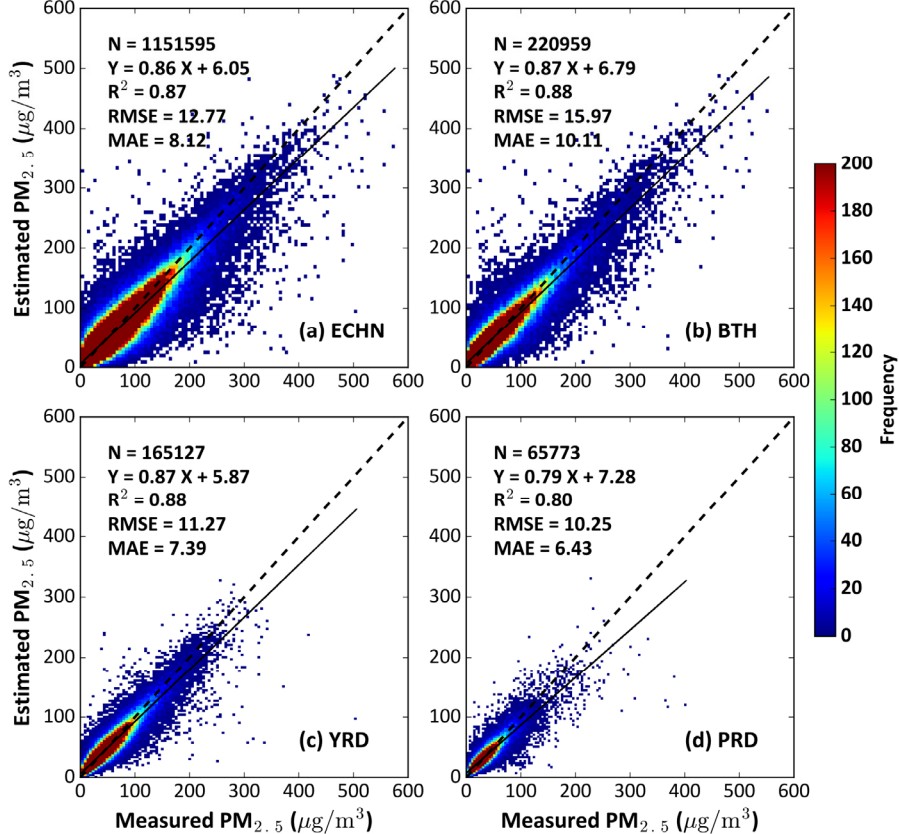

Figure 6. Density scatterplots of out-of-sample cross-validation results of hourly PM2.5 estimates (μg/m³) in 2018 for (a) eastern China, (b) the Beijing-Tianjin-Hebei (BTH) region, (c) the Yangtze River Delta (YRD), and (d) the Pearl River Delta (PRD) in China.


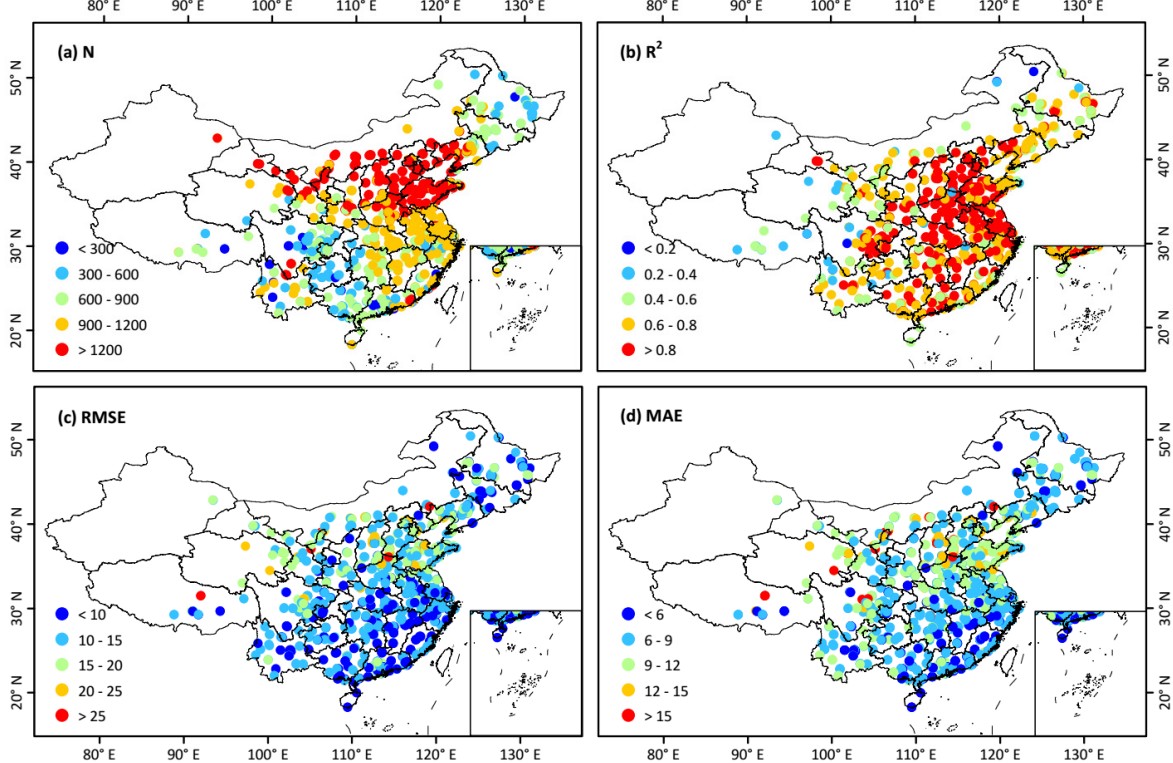

**Figure 7.** Individual-site-scale validation of hourly PM$_{2.5}$ estimates (μg/m$^3$) in 2018 in China in terms of (a) the sample size (N), (b) CV-R$^2$, (c) RMSE, and (d) MAE.

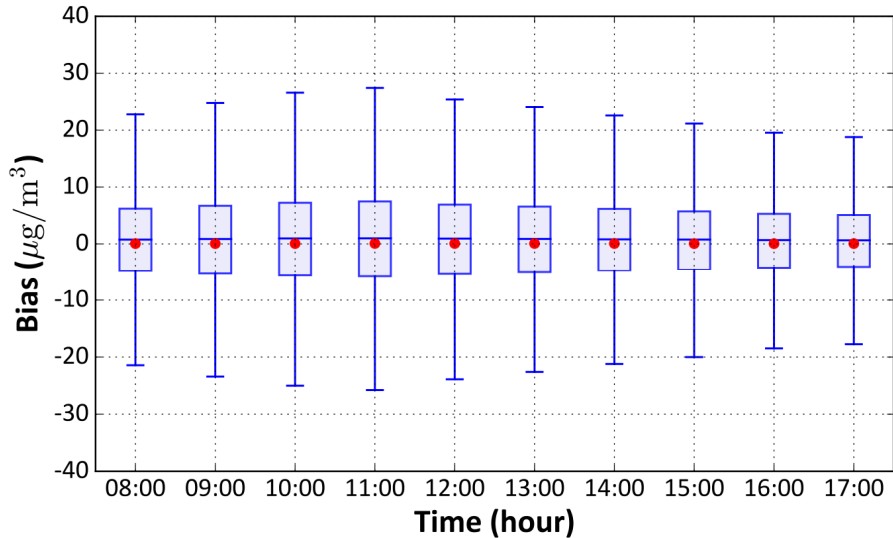

**Figure 8.** Boxplots of the temporal dependence of the bias in hourly PM₂.₅ estimates (µg/m³) in 2018 in China. In each box, the red dot represents the mean bias, and the blue middle, lower, and upper horizontal lines represent the median bias, 25$^{th}$ percentile, and 75$^{th}$ percentile, respectively.


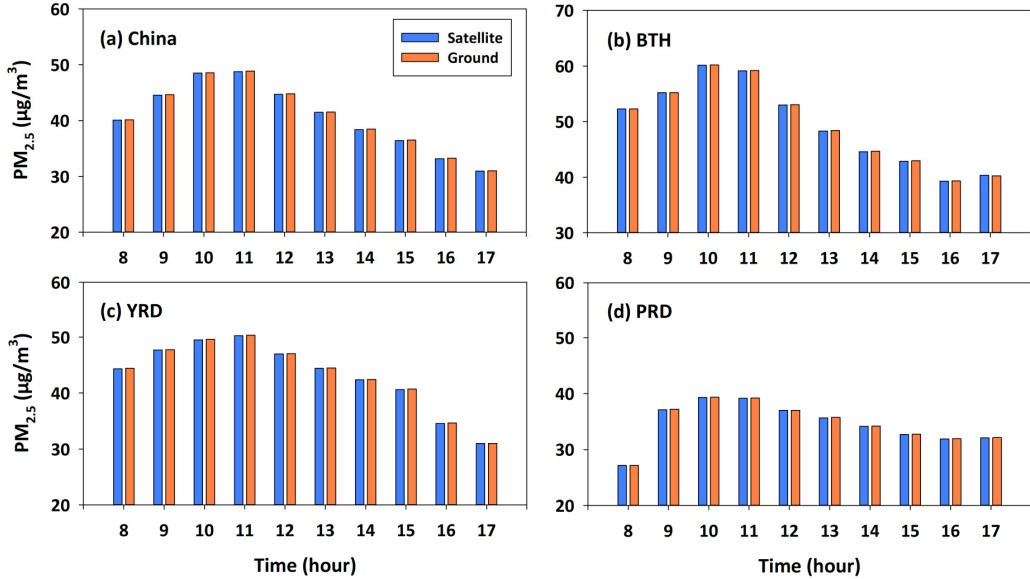

**Figure 9.** Time series of Himawari-8-derived (blue bars) and ground-based (orange bars) PM$_{2.5}$ diurnal variations (μg/m$^3$) in (a) China, (b) the Beijing-Tianjin-Hebei (BTH) region, (c) the Yangtze River Delta (YRD), and (d) the Pearl River Delta (PRD).


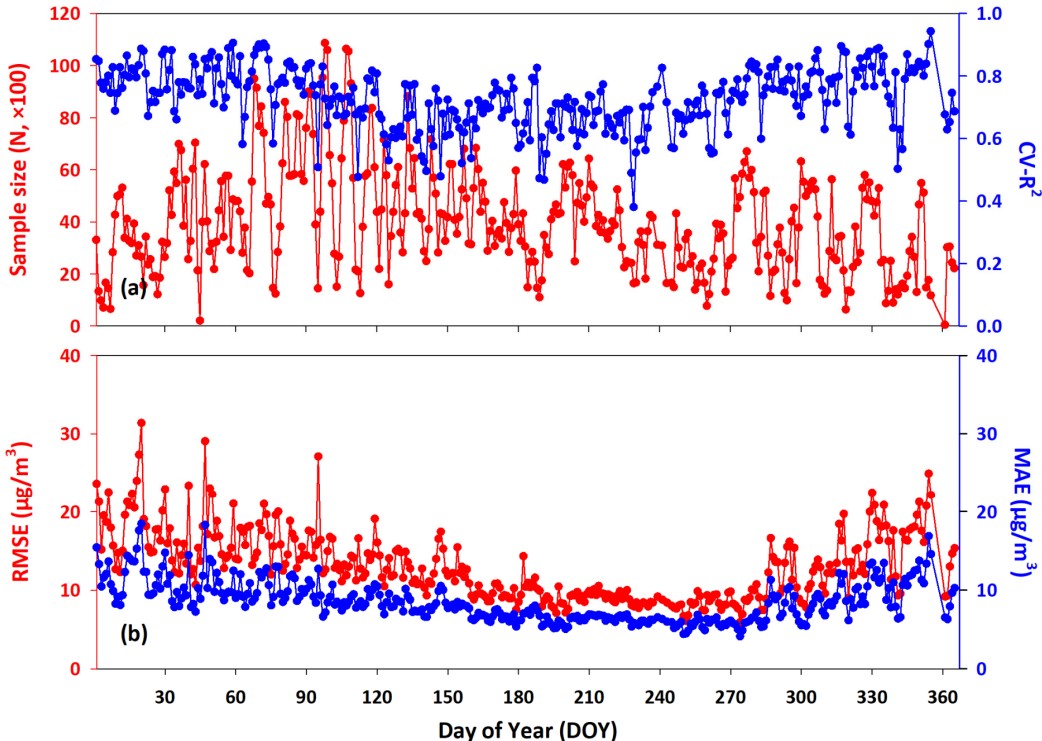

**Figure 10.** Time series of out-of-sample cross-validation of hourly PM$_{2.5}$ estimates (µg/m$^3$) in terms of (a) the sample size (N, red) and CV-R$^2$ (blue), and (b) RMSE (red) and MAE (blue) in 2018 in China.


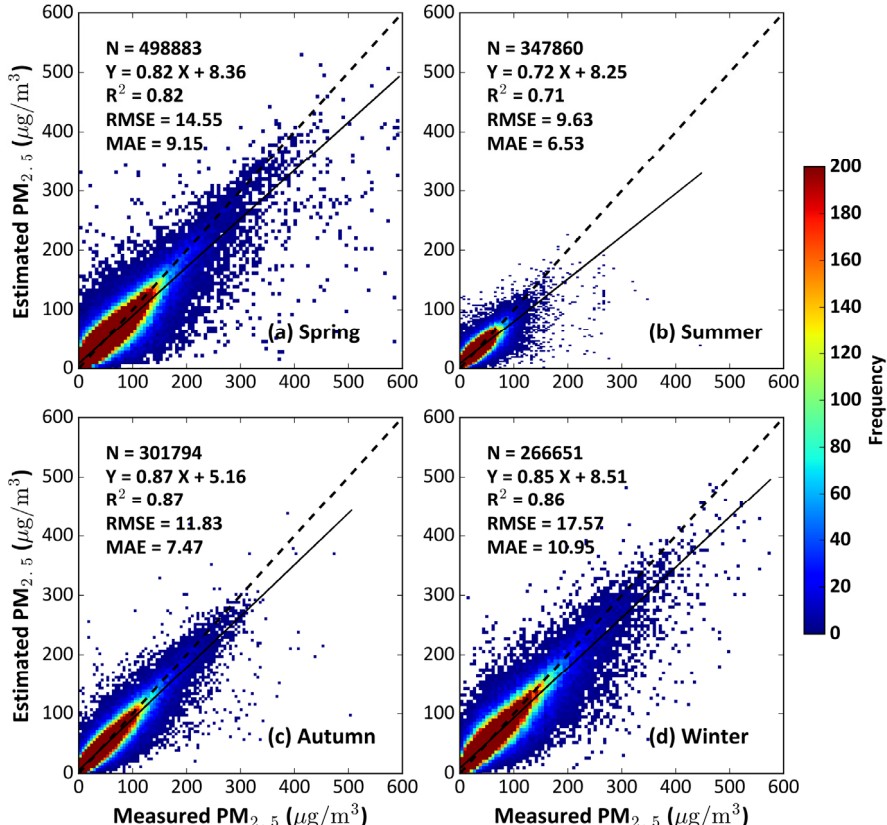

**Figure 11.** Density scatterplots of out-of-sample cross-validation results of hourly PM2.5 estimates (μg/m³) for (a) spring, (b) summer, (c) autumn, and (d) winter of 2018 in China. Dashed and solid lines denote 1:1 and best-fit lines from linear regression, respectively.


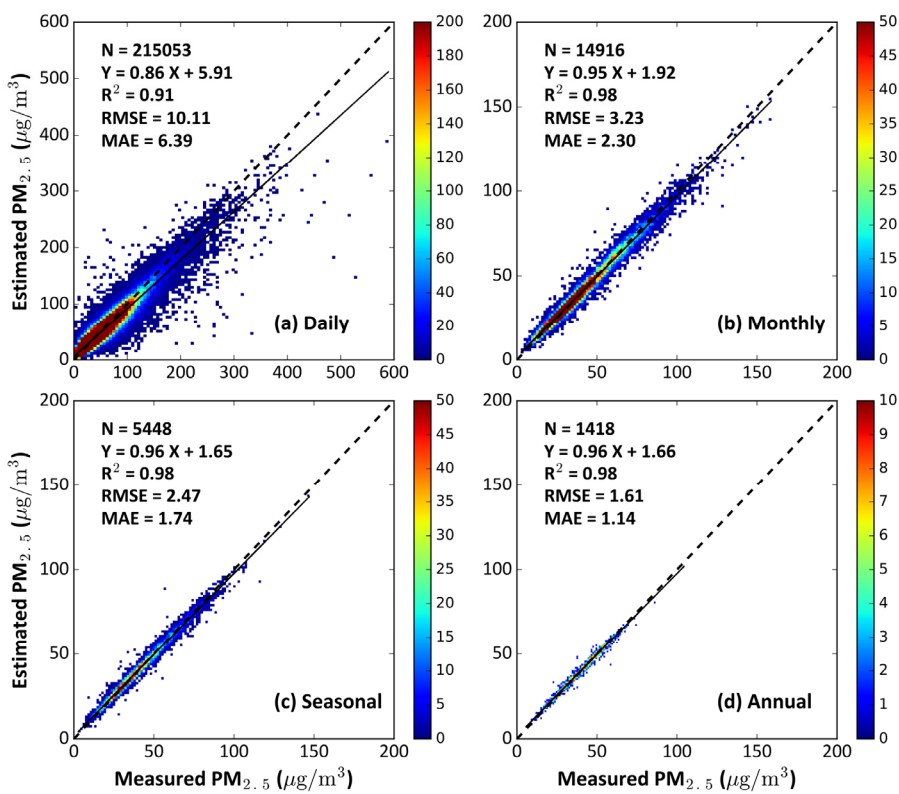

**Figure 12.** Density scatterplots of out-of-sample cross-validation results of (a) daily, (b) monthly, (c) seasonal, and (d) annual mean PM$_{2.5}$ estimates (μg/m$^3$) in 2018 across China.

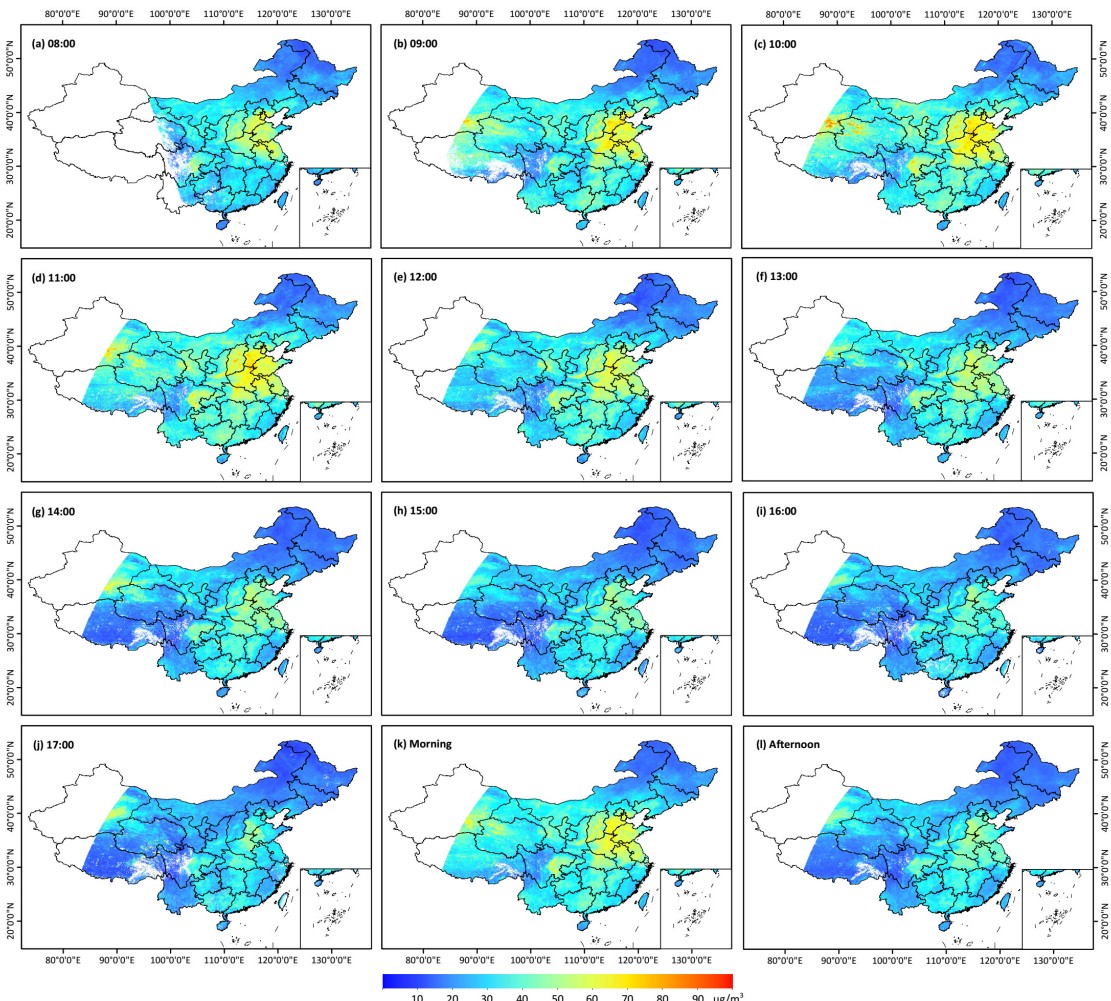


**Figure 13.** Himawari-8-derived hourly mean PM$_{2.5}$ maps (5 km) for different times of the day: (a) 08:00, (b) 09:00, (c) 10:00, (d) 11:00, (e) 12:00, (f) 13:00, (g) 14:00, (h) 15:00, (i) 16:00, (j) 17:00, (k) morning (08:00–12:00), and (l) afternoon (13:00–17:00) local time in 2018 across China.

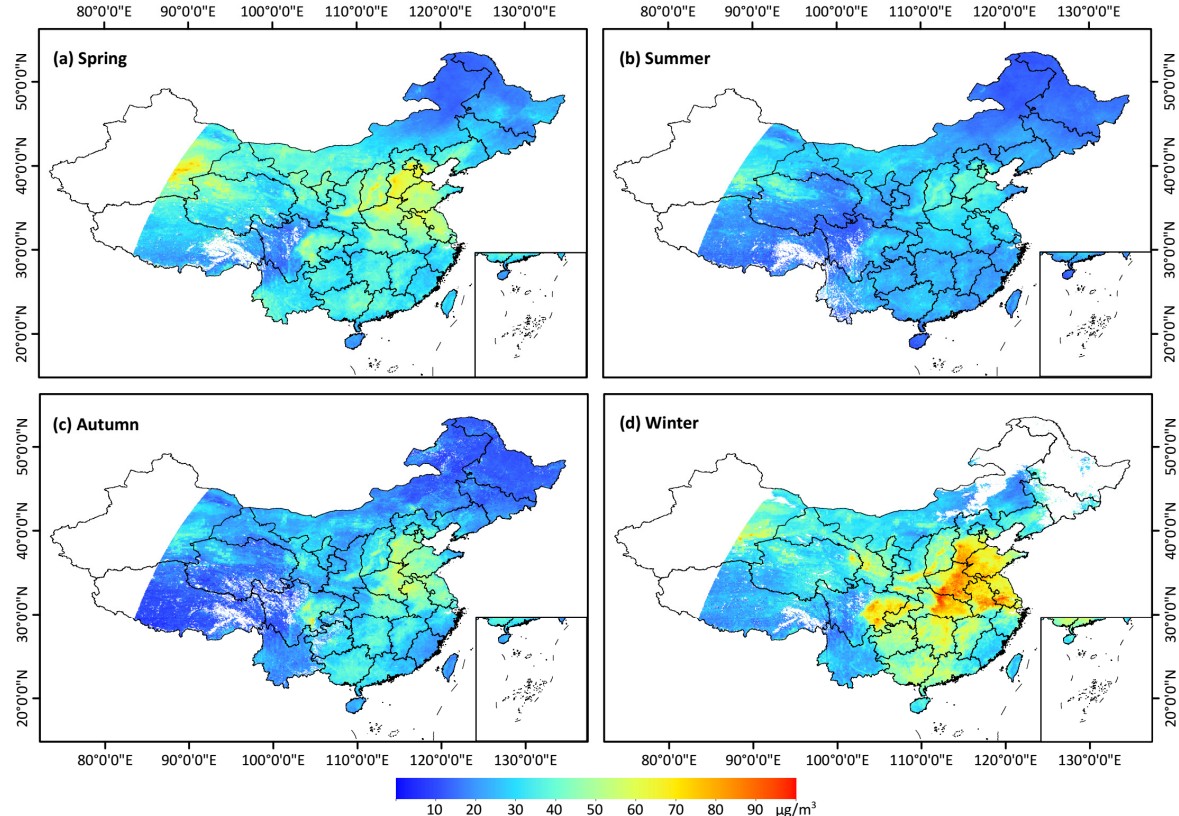


**Figure 14.** Himawari-8-derived seasonal mean PM$_{2.5}$ maps (5 km) for (a) spring, (b) summer, (c) autumn, and (d) winter of 2018 across China.

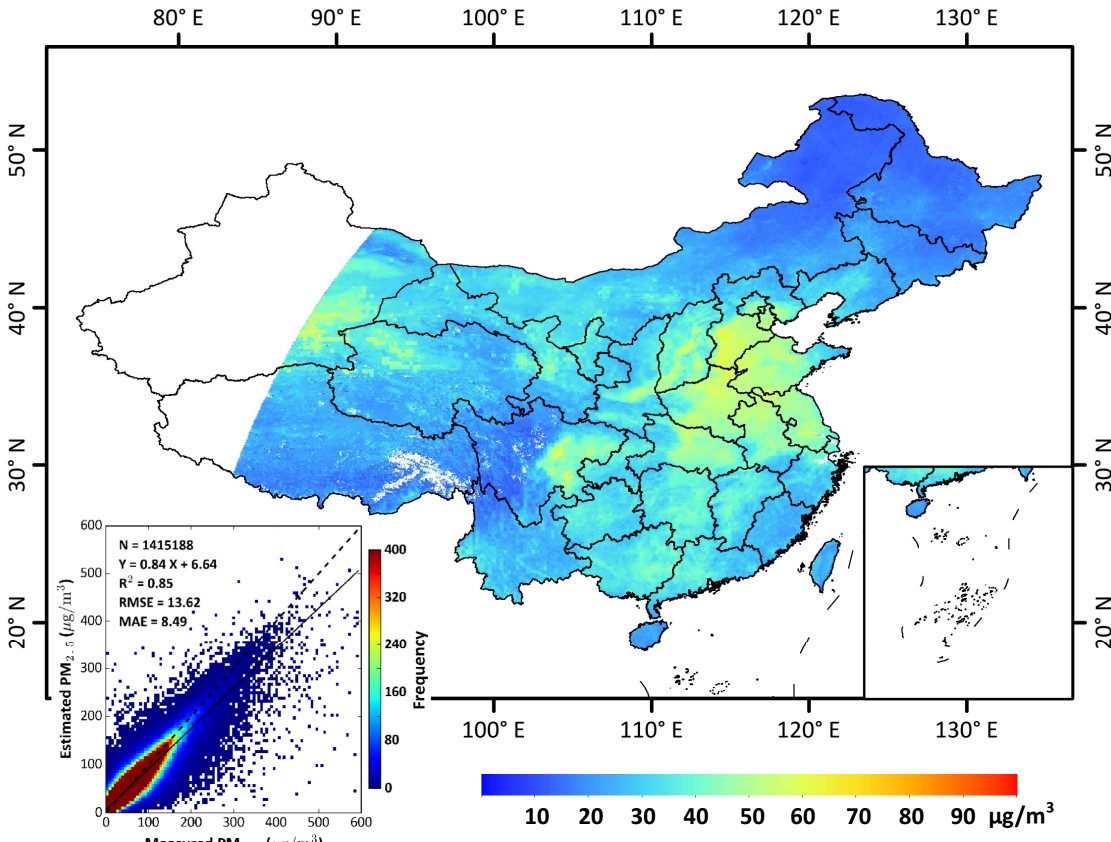

**Figure 15.** Himawari-8-derived annual mean PM2.5 map (5 km) for the year 2018 across China. The lower-left, inserted density scatterplot represents out-of-sample cross-validation results for all hourly PM2.5 estimates in China.