# Peer review of "Himawari-8-derived diurnal variations of ground-level PM2.5 pollution across China using the fast space-time Light Gradient Boosting Machine"

_Atmospheric Chemistry and Physics, 2020_

## Author Comment (AC1)

**Reviewer #1**

The authors estimated the ground-level PM2.5 pollution across China from Himawari-8 data by applying a fast space-time Light Gradient Boosting Machine. As an application study, authors investigated the diurnal variations of ground-level PM2.5 pollution. It was demonstrated that proposed application method is effective for accurate estimation of surface PM2.5.

**Response:** We appreciate the time and effort you have spent reviewing this manuscript; we have carefully revised it and provide responses to all the questions raised.

it is better to add relative reference (Zhang et al., 2020; Gui et al., 2020) in the introduction section and to describe the characteristics of this study.

**Response:** We have added these references and described the characteristics of our study in the revised Introduction.

It is better to extended the Section 2.2 into two parts, the first one citing the LightGBM model and its expansibility in PM applications, at the same time the authors should show what's the advantages of LightGBM in terms of other similar machine leaning models.

**Response:** We have split Section 2.2 into two parts, i.e., "2.2.1 LightGBM model" and "2.2.2 Model development". We have introduced the LightGBM model, its expansibility, and advantages in PM estimations in section 2.2.1 as follows:

"The LightGBM model, a newly developed tree-based machine-learning approach, which was introduced in 2017 (Ke et al., 2017). Using the gradient boosting framework to construct the decision tree, this approach can tackle both regression and classification tasks, and as such can be expanded for PM applications. It can also tackle the main challenge faced in traditional machine-learning approachesnamely, computational complexities, which are very time-consuming. LightGBM is a fast, distributed, and highly efficient method that reduces the number of data samples ($M$) and features ($N$).

…

In addition to the main technologies mentioned above, there are other features of the optimization, such as the leaf-wise tree growth strategy with depth restriction (Shi, 2007), histogram difference acceleration, sequential access gradient, and the support of category feature and parallel learning."

The second part address the modifications that the authors added or modified to the current model, detailing why these modifications are necessary and what kind of inspirations could be taken by readers to remote sensing or more broadly applications.

**Response:** We have clarified in the second part the modifications made and why we made them as follows:

"It is well known that air pollution has spatiotemporal heterogeneity, leading to large differences in PM$_{2.5}$ concentrations in both time and space. Such characteristics have always been ignored in most traditional statistical regression and artificial intelligence methods.

Studies have shown that including spatiotemporal information has led to improved PM$_{2.5}$ estimates using remote sensing techniques (Z. Li et al., 2017; Wei et al., 2019a, 2020). Therefore, we have introduced a new approach to integrate spatiotemporal information into the LightGBM model. The new model developed here is called the STLG model."

It is better to provide references for all models in Table 3.
**Response:** Done per your suggestion.

Which version of AOD is used from Himawari-8?
**Response:** The latest Himawari-8 Version 2 AOD product was employed. This has been clarified in the revised manuscript.

The caption of Figure 1 is not clear enough.
**Response:** We have expanded the caption of Figure 1 to make it clearer to the readers.

What would be the possible reason for the large gap in upper diagram in Figure 5, by the way, all the labels should be described in the caption.
**Response:** The large gap in the number of data samples is mainly caused by different degrees of cloud contamination in the satellite aerosol products for different days. We have clarified this in the revision. We have also added more descriptions of each panel in the caption for this figure.

Relative reference:
Zhang, T., He, W., Zheng, H., Cui, Y., Song, H., & Fu, S. (2020). Satellite-based ground PM2. 5 estimation using a gradient boosting decision tree. Chemosphere, 128801.
Gui, K., Che, H., Zeng, Z., Wang, Y., Zhai, S., Wang, Z., ... & Zhang, X. (2020). Construction of a virtual PM2. 5 observation network in China based on high-density surface meteorological observations using the Extreme Gradient Boosting model. Environment International, 141, 105801.

---

## Author Comment (AC2)

**Reviewer #2**

The authors developed a new space-time Light Gradient Boosting Machine (STLG) model for estimating ground-level PM2.5 concentration across China from Himawari-8/AHI AOD product and compared the hourly PM2.5 estimates with the ground measurements. The results demonstrated that the STLG model is more accurate than other tree-based machine learning models and previous studies as well as superior with faster learning speed and reduced memory consumption. It also suggests the importance of introducing spatio-temporal information into the development of the PM2.5-AOD relationship. However, some doubts remain about the reliability of the hourly PM2.5 estimates and their diurnal variation shown in this manuscript. I hope the authors will address the following major criticisms.

**Response:** We appreciate the time and effort this Reviewer has spent on this manuscript and the insightful comments. We have carefully revised our manuscript according to your suggestions, and the responses to the criticisms raised in your report.

Major Criticisms
In addition to the Himawari-8 AOD, many other variables shown in Figure S1 were used in the STLG model as features to estimate ground-level PM2.5 pollution. However, the manuscript does not discuss the contribution of each variable to the estimation. There should be a quantitative discussion of which variables contribute to PM2.5 estimation and to what extent. For example, it may be useful to calculate the importance of each variable in the model, or to compare the RMSEs for different combination of feature variables.

**Response:** We have calculated the importance of each variable of the model in estimating ground-level $PM_{2.5}$ concentrations and added the discussions in the revision according to your suggestions. It reads as follows:

"In addition to Himawari-8 AODs, other auxiliary variables were considered and employed to improve $PM_{2.5}$-AOD relationships. However, to avoid redundant information, we first calculated the normalized importance (%) of each feature to the $PM_{2.5}$ estimation during the model training (Figure 2). It represents the total gains of splits that use the feature during the decision-tree construction, but not the physical contribution. AOD is found to be the most important feature, accounting for about 17%. All meteorological factors have an important impact on the $PM_{2.5}$ estimation, especially BLH, RH, and TEM (importance > 8%). Followed by two surface-related variables (i.e., NDVI and DEM) and POP. Finally, the influence of aerosol precursors and emissions (i.e., $NH_3$, $NO_x$, $SO_2$, PM, and VOC) on the $PM_{2.5}$ estimation cannot be ignored (importance > 2%). Therefore, all 16 selected variables are included to establish the final model in this study."

[Figure]

Figure 2. Sorted normalized importance (%) of each feature in the PM$_{2.5}$ estimation during the model construction.

Previous studies have showed the time-dependent bias in the Himawari-8 AOD product used in this study (e.g., Wei et al., 2019b). It is therefore preferable to quantify the temporal dependence of bias in the hourly PM2.5 estimation before discussing the diurnal variations of PM2.5. The manuscript seems to show the temporal dependence of the RMSE (e.g., Figure 2 and Table 1), but not the temporal dependence of the bias.

**Response:** We have quantified the temporal dependence of the bias in hourly PM$_{2.5}$ estimates and added discussions in the revision according to your suggestions as follows:

"We first quantified the time series of the bias in hourly PM$_{2.5}$ estimates during the day in China (Figure 8). There is a slight temporal dependence, where the PM$_{2.5}$ bias increases gradually with increasing standard deviation, reaching a maximum around 11:00 a.m., and subsequently decreasing. This seems to be closely related to the diurnal variation of PM$_{2.5}$ concentrations. The PM$_{2.5}$ estimates are less affected by the time-dependent bias in the Himawari-8 AOD product (Wei et al., 2019b) because machine learning is not sensitive to the systematic bias of aerosol retrievals (Wei et al., 2021b). Nevertheless, our model is generally robust, and can accurately estimate PM$_{2.5}$ concentrations with small mean (median) biases of 0.05–0.08 (0.63–0.99) µg/m$^3$ during different hours throughout the day."

[Figure]

**Figure 8.** Boxplots of the temporal dependence of the bias in hourly PM2.5 estimates (μg/m³) in 2018 in China. In each box, the red dot represents the mean bias, and the blue middle, lower, and upper horizontal lines represent the median bias, 25th percentile, and 75th percentile, respectively.

In Table 1, the hourly mean concentrations of PM2.5 reach their maximum at 10 a.m., whereas they are lowered at sunrise or sunset. However, the RMSE of PM2.5 estimates is as large as the magnitude of these diurnal variations, and the RMSE appears to be proportional to the value of PM2.5. I believe the authors should ensure that the diurnal variations of PM2.5 derived from Himawari-8 are reasonable compared to ground-based measurements.
**Response:** Thanks for your suggestion. In addition to the temporal dependence of the bias in hourly PM2.5 estimates (see the above comment), we have also compared Himawari-8-derived and ground-based PM2.5 diurnal variations from all available monitoring stations at both national and regional scales in China as follows:

"We also compared Himawari-8-derived and ground-based PM2.5 diurnal variations from all available monitoring stations in China and three typical urban clusters (Figure 9). Hourly PM2.5 concentrations observed by satellite are highly consistent with ground-based measurements, with a small difference within ±0.10, 0.11, 0.13, and 0.11 μg/m³ in China and in each region, respectively. Moreover, the same diurnal variations of PM2.5 pollution are seen during the day, i.e., they reach their maximum values at 10:00 or 11:00 and are lower at sunrise and sunset. These results illustrate that the diurnal PM2.5 variations derived from Himawari-8 are reasonable compared to ground-based measurements."

[Figure]

**Figure 9.** Time series of Himawari-8-derived (blue bars) and ground-based (orange bars) PM$_{2.5}$ diurnal variations (μg/m$^3$) in (a) China, (b) the Beijing-Tianjin-Hebei (BTH) region, (c) the Yangtze River Delta (YRD), and (d) the Pearl River Delta (PRD).

Minor Criticisms

Please spell out the abbreviations: MISR, MODIS and VIIRS in Lines 49-50; RMSE and MAE in Line 92; NDVI, STRM and DEM in Lines 114; MEIC in Line 107;

**Response:** We have spelled out all abbreviations in the revised manuscript.

Line 37: 'Zhang et al., 2017' is missing from References.

**Response:** It should be "Q. Zhang et al., 2019" here. This has been corrected.

Line 39: 'Sun et al., 2014' is missing from References.

**Response:** It should be "Sun et al., 2004" here. This has been corrected.

Line 72: Which dose 'Zhang et al., 2019' refer to 'Zhang, Q. et al., 2019' or 'Zhang, T. et al., 2019' in Reference?

**Response:** It should be "T. Zhang et al., 2019" here. We have added first-name initials where needed in the manuscript.

Line 159: 'Rodriguez et al., 2010' is missing from References.

**Response:** We have added it to the reference list.

Line 209: '0' should be removed.

**Response:** Done per your suggestion.

Lines 226: Please explain 'a harsher environment and more intense human activities' in more detail.

**Response:** In the revision, we have rewritten this as "due to the harsher environmental conditions (e.g., low humidity and less precipitation) and more intense human activities then (e.g., coal heating and straw burning) in winter and spring".

(Tables)
Table 1: Is the uncertainty '±49.31' at BTH/9:00 correct value? It seems to be abnormally large compared to the others.
**Response:** This was a typo. The actual uncertainty is "±15.04". We have corrected this in the revised manuscript.

(Figures)
Labels for y-axis in Figures 2, 3, 6, S2, S3 and S4: The authors labeled the y-axis of Figures 2, 3, 6, S2, S3 and S4 with different labels such as "Model-fitted PM2.5", "Station-based-CV PM2.5", "Model-CV PM2.5" and "Estimated PM2.5". However, some difference may exist, but they should all represent the PM2.5 estimated by the STLG model. If so, I believe it would be clearer to use the same label, such as "Estimated PM2.5".
**Response:** We have rewritten the labels of the y-axes in these figures as "Estimated $PM_{2.5}$" according to your suggestion.

Captions of Figures 2, 3, 4, 5, 8, S2, S3 and S4: Although Figures 2, 3, 4, 5, 8, S2, S3 and S4 have several panels, explanations are missing from the captions. Please add explanations of each panel to the captions, as shown in Figures 6 and 7.
**Response:** We have added explanations to each panel in the caption for all the required Figures in the revision, according to your suggestions.

(Supporting Information)
Tables and Figures in the "Supporting Information" section: I wonder why the authors placed so many tables and figures in the " Supporting Information" section. They seem to be part of the main results of this study and used in the discussion demonstrating the advantages of the proposed STLB model. Unless the authors have a clear reason, it would be preferable to place these tables and figures in the full text of the manuscript.
**Response:** Thanks for your suggestion. We have moved all tables and figures in the Supplement File to the main text in the revised manuscript.